# The Stress of Competing: Cortisol and Amylase Response to Training and Competition

**DOI:** 10.3390/jfmk6010005

**Published:** 2021-01-04

**Authors:** Roberta De Pero, Carlo Minganti, Giuseppe Cibelli, Cristina Cortis, Maria Francesca Piacentini

**Affiliations:** 1Department of Movement, Human and Health Sciences, University of Rome “Foro Italico”, 00135 Rome, Italy; roberta.depero@uniroma4.it (R.D.P.); carlo.minganti@uniroma4.it (C.M.); 2Department of Biomedical Sciences, University of Foggia, 71122 Foggia, Italy; giuseppe.cibelli@unifg.it; 3Department of Human Sciences, Society and Health, University of Cassino and Lazio Meridionale, 03043 Cassino, Italy; c.cortis@unicas.it; 4Department of Human Physiology and Sportsmedicine, Vrije Universiteit Brussel, 1040 Bruxelles, Belgium

**Keywords:** intensive training, TeamGym, stress response, well-trained athletes

## Abstract

TeamGym is a popular form of gymnastics, including tumbling (Tu), trampette (Tr) and floor exercises (F) characterized by intensive practice placing high levels of stress on athletes. The aim of the study was to investigate athletes’ stress-related changes during TeamGym training and competition, considering hormonal and enzymatic responses (i.e., salivary cortisol and alpha-amylase). Ten (5 males and 5 females) TeamGym athletes (age: 22–28 y) were tested twice at the same time before training and competition; furthermore, for excluding circadian effect on hormonal and enzymatic responses, they were tested at the same time during a rest day. Alpha-amylase and cortisol were measured 15 min before the beginning of exercise, after each gymnastic equipment performance, and after thirty minutes from the end of the performance. Factorial ANOVA with repeated measures was used to verify differences between training and competition (*p* < 0.05). Competition elicited higher values of alpha-amylase than training (*p* ranging from 0.001 to 0.019) and rest (*p* ranging from 0.001 to 0.019). Cortisol showed no exercise induced increase, and its concentrations were higher prior to training compared to competition. TeamGym responses confirm other sports findings in stating that competition elicits higher stress response than training and suggest that salivary alpha-amylase is a more sensitive marker than cortisol to psychophysiological stress also in gymnastics intermittent performance.

## 1. Introduction

Originating from Scandinavia in 1996, TeamGym is a popular form of gymnastics with 6–12 members in female, male or mixed teams performing three different events: trampette, tumbling and floor exercise [1]. Trampette and tumbling program consist of three different rounds of series of acrobatic elements to be accompanied by music. Only six members from the team are chosen for each round and do not have to necessarily be the same athletes every round; they must perform acrobatic elements consecutively and close to each other. The floor program consists of a 3 min choreographic exercise (jumps/leaps, balances, body waves) simultaneously performed by all the team members. The team receives a total score for each event. In TeamGym, like in gymnastics, the technical difficulties of the required skills, the nature of the event, and the competition level can pose a high physical and psychological load on athletes [2,3]. The athletes’ capability to deal with environmental demands is a crucial factor [4] and could be considered one of the parameters for selecting talented athletes and a useful predictor of gymnastics performance [5]. Athletes’ subjective responses to stressful and demanding competitive settings are dynamic and complex [6], and to accurately describe the competition stress-related responses of athletes, multiple variables should be assessed during authentic competitive situations, especially during different official level events [2,7]. On this basis, the activity of the hypothalamic–pituitary–adrenocortical system (HPA) with the secretion of cortisol, and the sympathetic adrenomedullary system (SAM), with the secretion of alpha-amylase, have been largely used as objective markers of psychophysiological performance stress [7,8,9,10] in different disciplines [2,3,11,12,13,14]. Several studies suggested that salivary alpha-amylase (sA-A) increases more rapidly than salivary cortisol (sC) to a stressor suggesting that it is a more immediate indicator of stress than cortisol [15,16,17]. Presumably, the differences in response to stress reaction between sA-A and sC may result from the differences in the time latency between the stress response of the sympathetic nervous system (SNS) and that of the HPA axis [17]. Furthermore, while SNS activity increases in response to challenges that are perceived as manageable or controllable, HPA axis response is more likely during emotionally distressful or uncontrollable situations [7]. Thus, the examination of both independent and interactive effects of the SNS and the HPA axis in the context of competition may represent the key in advancing the understanding of the role of physiological responses to stress [7].

In order to be able to measure the stress response to training and competition in ecological settings, noninvasive methods need to be used [7,8,9,10]. It has been demonstrated that challenges associated with training and competition settings can induce distinct responses; in fact, competition is characterized by performance (i.e., competition importance, spectator or judge’s evaluation, opponent ability) and organizational stressors (competition format, performance advancement) [3] that are not present during training. The higher stress response associated with competition compared to practice or laboratory has emerged in a variety of sports such as during soccer and Jiu-Jitsu [13,18], in running [19], and in golf performance [20]. Only one study [21] demonstrated greater heart rate and cortisol responses during competition compared to training in gymnastics, but the results can be compared with some cautions to TeamGym performance: first of all, the findings of Filaire and colleagues [21] concern peripubertal female gymnasts and describe their psychological and physical stress induced by training or competition which lasts 90 s of continuous effort. No studies, to our knowledge, evaluated adult male and female gymnasts during training and official TeamGym competition [2,15] that lasts longer than gymnastic competitions. Moreover, the intermittent effort required by TeamGym athletes is performed on different sports equipment. Furthermore, the TeamGym training method has not yet been well established, mainly adapted from artistic gymnastics not always considering the technical and functional different performance demands.

Thus, it is reasonable to assume that the understanding of TeamGym responses in both settings will provide valuable data regarding the magnitude of applied training loads and performance improvement [22].

Thus, the aim of this study was to apply an integrated ecological physiological measurement of stress in adult gymnasts during training and the most important TeamGym competition (European Championship). Considering the higher distressful and incompletely controllable situations of performing high difficulty acrobatic elements during competition compared to training [7], we hypothesized that the two major systems involved in responses to stress (i.e., SAM and HPA) would react differently to these two situations and that TeamGym European competition would impose a greater psychophysiological load on athletes when compared to training.

## 2. Materials and Methods

### 2.1. Participants

Ten elite TeamGym athletes (5 males and 5 females) participated in this study and provided a written informed consent form. They were selected gymnasts for representing the Senior Italian mixed team at the European TeamGym Championship and had at least 5 years of previous training (consisting of 2.5-h sessions, three times per week). All participant’s anthropometric and physiological measures are represented in Table 1.

### 2.2. Design

The local ethical committee approved the protocol employed (Protocol 2523/15 November 2012). Three experimental sessions (i.e., training, competition, and a rest day) were scheduled. In particular, the experimental training session was planned ten days before the competition at the end of a highly intensive pre-competitive period, consisting of two sessions each lasting two hours focusing on the refinement of routines for competition; the rest data were collected after one week of full recovery. To evaluate the bio-humoral responses to stress, sA-A and sC were measured 15 min prior to the beginning of the competition (pre-competition), immediately after the end of each apparatus (post-tumbling, post-trampette, post-floor) performance (scheduled with a 30-min cadence), and at 10- and 30-min post-competition recovery phase. For more details, see De Pero and colleagues [2]. In considering that the European team competition was scheduled between 1200 h and 1400 h, to avoid effects of circadian variations in sC [10] and sA-A [23], the same time schedule was used to collect salivary samples both during the training session and during the recovery day. Specifically, after having received the time schedule of the competition, the training session was planned not only at the same time of the day but also with the same tournament and with the same rest period between apparatus as the competition program. Therefore, the area under the curve (AUC) for sC and sA-A during training (i.e., sCAUC-training and sAAAUC-training), competition (i.e., sCAUC-competition and sAAAUC-competition) and during time-matched rest day (i.e., sCAUC-rest and sAAAUC-rest) were calculated using the trapezoidal system.

### 2.3. Methodology

#### 2.3.1. Saliva Collection and Assays

Salivette sampling devices (Salivette, Sarstedt, Germany) were used to obtain saliva samples (>0.05 µL), and gymnasts were instructed to place the cotton swab into their mouth under their tongue for 2 min and to chew 20 times. They were instructed not to swallow during this sampling period. The swab was then returned to the Salivette tube, which was immediately frozen for storage at −30 °C until the time of assay. Athletes were fasted since breakfast (around seven o’clock), and they were asked not to rinse their mouth with water prior to sampling. Then, after ascertainment of salivary blood contamination absence, saliva samples were centrifuged at 3000× *g* rev.min^−1^ for 15 min at 4 °C, stored at −80 °C, and assayed in the same series to avoid inter-test variations. An enzyme immunoassay kit was used to measure sC concentrations, and a kinetic reaction assay kit was used for sA-A measurements, respectively (Salimetrics LLC, State College, PA, USA), according to manufacturers’ instructions. Cortisol intraassay coefficient of variation of 3.5 ± 0.5% and inter-assay reproducibility of 5.08 ± 1.33% were accepted. Amylase intraassay coefficient of variation and inter-assay reproducibility of 5.47 ± 1.49% and 4.7 ± 0.15% were accepted. A standard plate reader (Power Wave XS, Bio-Tech Instruments, Winooski, Vermont, USA) was used for salivary determination by 450 nm and 405 nm filters for sC and sA-A, respectively was used.

#### 2.3.2. Statistical Analysis

The statistical package IBM SPSS version 20 (IBM, Chicago, IL, USA) was used for the analysis. All data are expressed as means ± SD. The Shapiro-Wilk test was applied before the analysis to test the normal distribution of the data. For sC and sA-A, two separate factorial ANOVA with repeated measures were used (condition ×3 [competition, training and rest matched time] and time × 5 [pre, post tumbling, post-trampette, post floor, and post 30′]). When significant interaction (i.e., condition for time) was observed, follow-up tests were conducted running separate repeated-measures ANOVA for conditions (i.e., competition, training and rest matched time) to explore the different effects of time in the three conditions. Salivary cortisol AUC and salivary alpha-amylase AUC were analyzed by means of two separate ANOVA with repeated measures (condition × 3). Before the analysis, Mauchly’s test for sphericity was applied to control for statistical assumptions, and if sphericity had been violated, Greenhouse–Geisser degrees of freedom correction factor was used. Post hoc comparisons were performed by means of Fisher’s least significant difference test, and the Bonferroni alpha level correction was applied. The significance level for all comparisons was set at *p* ≤ 0.05. In addition, effect size (ES) was calculated for all variables as partial eta-squared (η^2^*p*). Partial eta-squared values below 0.01, between 0.01 and 0.06, between 0.06 and 0.14, and above 0.14 were considered to have trivial, small, medium, and large ES, respectively [24].

## 3. Results

For all the variables, no outliers or non-normal distribution were detected.

SA-A values showed a significant main effect for condition (F(2;12) = 83.48; *p* < 0.001; η^2^*p* = 0.933), time (F(4;24) = 45.55; *p* < 0.001; η^2^*p* = 0.884) and for the interaction effect for condition by time F(8;48) = 7.08; *p* < 0.001; η^2^*p* = 0.541). Follow-up analysis showed significant higher values in the pre competition with respect to the rest and training matched time (*p* range between 0.001 and 0.023) and a significant increase during competition with a return at lower values, with respect to pre (*p* range 0.001 to 0.0023). SA-A training matched time condition post floor was different from pre (*p* < 0.001) while no differences were found between the other time points. No differences were found in the five time points for the rest matched time condition.

The SC values showed a significant main effect for time (F(4;24) = 8.67; *p* < 0.001; η^2^*p* = 0.591) and for the interaction effect for condition by time (F(8;48) = 7.08; *p* < 0.001; η^2^*p* = 0.541). Follow-up analysis highlights lower values in the pre-competition with respect to rest (*p* = 0.032) and training matched time points (*p* < 0.001). No significant variation over time in and between the three conditions was found.

No differences emerged between sCAUC-competition (182.75 ± 7.86 au), sCAUC-rest (152.87 ± 22.99 au) and sCAUC-training (155.62 ± 8.68 au). sAAAUC showed a significant main effect for condition (F(2;18) = 184.97; *p* < 0.001; η^2^*p* = 0.969). In the post hoc analysis differences (*p* = 0.001) were found between sAAAUC-competition (676.19 ± 162.78 au), sAAAUC-rest (213.34 ±168.33 au) and sAAAUC-training (342.44 ± 189.99 au) (Figure 1 and Figure 2).

## 4. Discussion

The main findings of this study were that the SAM and the HPA axis reacted differently to stressful demands of training and competition and that sA-A can provide valuable information for coaches regarding athletes’ responses to training and competition. In fact, no significant cortisol increases during training nor competition emerged; cortisol was significantly higher prior to training compared to the competition and the rest matched the time sample. However, this finding is not surprising, considering that the training session was scheduled after returning from a 10-day training camp. Increases in basal cortisol concentrations after periods of intensified training have already been reported in the literature [25]. However, findings are inconsistent because this increase seems to depend on the training status of the athlete. Decroix and colleagues [26], in fact, showed no effect of seven days of intensified training on basal cortisol in well-trained cyclists. These athletes exhibit the required mechanisms to cope with the stress of intensive training, preventing intensive training induced changes in cortisol. On the contrary, our athletes, although highly trained, were not accustomed to training camps and most probably were less able to tolerate a 4-fold increase in training volume. As expected, we found no anticipatory cortisol response to both training and competition, confirming that experienced and well-trained gymnasts are able to control competitive stress; several authors [7] have already speculated that cortisol secretion is more likely to increase during emotionally distressful or uncontrollable situations. Furthermore, the literature states that cortisol response to physical activity is dependent on the intensity [10] and the duration of activity [14]. TeamGym competition is characterized by intermittent bursts of activity interspersed with equivalent or longer periods of rest and recovery; thus, it was plausible to speculate, as already seen for peripubertal gymnasts, that either the program was not intense enough to stimulate additional cortisol secretion, or that the gymnasts were well adapted to the training loads encountered [27]. On the contrary, significant increases of sA-A levels prior to and during competition with respect to training and baseline concentrations emerged, as previously observed in competitive settings [3,7,8,23,28]. Alpha-amylase response to exercise is more rapid and acute than that of cortisol [29]; the sA-A increase before training and competition could support the activation in preparation for performance and suggests that sA-A can be modulated by emotional and affective states associated with an official competition [30]. Furthermore, these results are in line with those reported by several authors [7,11,29]; in particular, sA-A showed a fast response with peak values at the end of training and competition and full recovery after 30 min. It is likely that the rapid decrease in alpha-amylase following cessation of exercise may be due to the intermittent nature and to the short duration of the activity [11]. Furthermore, this study confirms the assumption that real competition can amplify the stress responses when compared with training sessions [13,21,22]: in fact, sA-A secretion during the competition was significantly higher than during training. Although practice itself can be quite stressful, real competition appears to promote greater responses, probably because of the additional stressors such as novelty, unpredictability, the importance of the event, public display and judgment of the skills [3]. Thus, a competitive environment can amplify the stress imposed on the athletes because of additional psychological and physical demands. The main aim of physiological assessment during real competition is to identify determinants of performance, profile athletes, and provide support for training program validity. In addition, psychological factors should be investigated before the events to improve the knowledge about the way that TeamGym athletes cope with the stress inherent to official competitions [13]. Identifying the psychophysiological determinants of performance is crucial not only for predicting performance itself but also for profiling athletes and for prescribing training in the effort of translating observations into training prescription. To achieve specific adaptations, a training program must stress the systems that are exactly engaged during competition [22]; thus, TeamGym coaches should aim to mimic competition demands during training sessions assessing internal stressors imposed by competition settings. TeamGym, like gymnastics, depends on the perfect execution of specific movements. Therefore, the adaptation of the athlete to a different type of stress can increase overall performance. For example, coaches can implement training sessions by simulating competition with the same competition rest periods among trials, with athletes from other clubs or teams, or with the presence of external judges.

## 5. Conclusions

Competition is a stressful situation, which stimulates high psychophysiological responses in athletes [2,21]. The present findings not only provide valuable information on the difference in temporary changes in stress-related parameters during real competition and training but also give indications on the chronic changes that occur after intensive training in well-trained gymnasts. Moreover, higher stress-related responses during competitions have been shown to be related to higher penalties during TeamGym [2], and this occurrence could potentially increase the risk of injuries. Quantifying the psychophysiological response to exercise can provide a wealth of information in planning a progressive adaptation during training to the high demands of competition and may provide another tool to help enhance athletes’ performance. As the small sample size, this study can be considered only a preliminary investigation: in the future, in order to be complete, more information regarding psychophysiological load (i.e., heart rate and blood lactate responses), mood changes [14] and anxiety [2] before training and competition could help coaches and physicians to individualize training and avoid non-functional overreaching.

## Figures and Tables

**Figure 1 jfmk-06-00005-f001:**
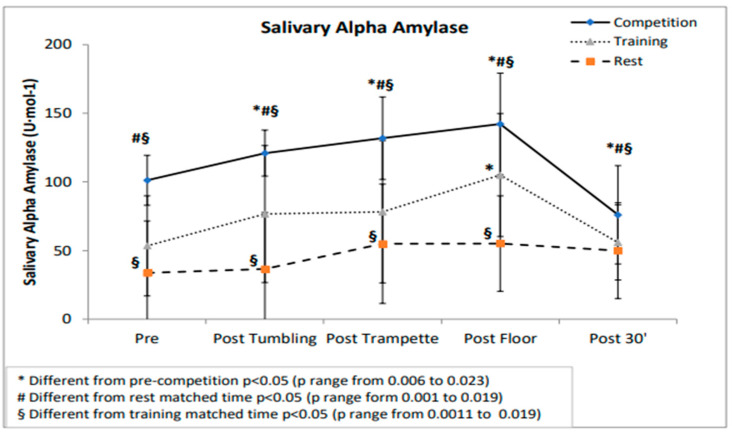
Salivary alpha-amylase values during the competition, training ad rest day.

**Figure 2 jfmk-06-00005-f002:**
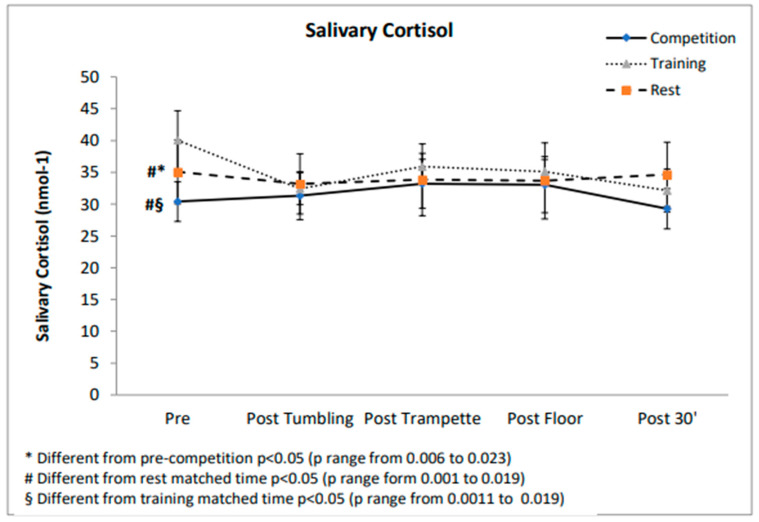
Salivary cortisol values during competition, training and rest day.

**Table 1 jfmk-06-00005-t001:** Means (±SD) of anthropometric and physiological measures of TeamGym athletes.

	Age (yrs)	Body Mass (kg)	Height (cm)	BMI (kg/m^2^)	VO_2_peak (ml/kg/min)	HRmax (BPM)
**Males**	27 ± 2	65 ± 4	171 ± 3	23 ± 1	50 ± 6	195 ± 6
**Females**	24 ± 2	55 ± 3	162 ± 3	21 ± 1	43 ± 7	193 ± 7

## Data Availability

Data is contained within the article or supplementary material.

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
