# Peer review of "The Stress of Competing: Cortisol and Amylase Response to Training and Competition"

_jfmk, 2021, doi:10.3390/jfmk6010005_

Round 1

Reviewer 1 Report

Review
The aim of this research was to examine the sympathetic adrenal medullary (via salivary alpha amylase measurement) and hypothalamic pituitary adrenal axis responses to a TeamGym training session and competitive setting.
TeamGym is a team game of 3 different events: Trampette; tumbling and floor exercises. This study is to examine some stress responses to an official TeamGym event. The variables measured are formed from the HPA and SAM systems.
Introduction
This is a well written introduction and leads the reader easily towards the research aim. There are a few comments that I would suggest are addressed to strengthen this section.
Page 2
Line 48: Alpha amylase is not a catecholamine. It is an indicator of the sympathetic adrenal medullary activity and has been substituted for catecholamine measures. Also “i.e” should be replaced with “i.e.”
Line 51: Remove capital “C” from “cortisol”.
Line 52: remove space between “16, 17”.
Line 52: This is simply due to the mechanism of synthesise of cortisol. Perhaps it would be appropriate to state that it is a more immediate indicator of stress reactivity compared with salivary cortisol.
Line 54: define at first use of “SNS”.
Materials and Methods
Page 2
Line 84: The title of this section is participants. Perhaps maintain this terminology instead of “subject”.
Page 3
Line 106: Alter to the Salivette Cortisol. It is assumed this was the type of collection tube used?
It is known that the Salivette cotton swab can influence the cortisol measures.
Line 107: How long was the fast, the session was 12:00 – 14:00 so were they fasted after a breakfast meal or was if from the previous night.
Line 107: Can you confirm that each trial for each participant was completed at the same time of day i.e. if I completed the first trial at 13:00 all the other trials were at this time of the day.
Line 126: Should this be Greenhouse-Geisser?
Results
Page 4
Line 147: It is not normally practice to present nanomoles as “nm”. I would alter these to nmol.
Line 147: The cortisol concentrations seem quite high. At the time of day I’d expect values of 5 – 15 nmol.L-1.
Figure 1: Indicate where the statistical differences were found
Discussion
This is a well written section.

Author Response

Answer to Reviewer #1:

Page 2

  • Line 48: Alpha amylase is not a catecholamine. It is an indicator of the sympathetic adrenal medullary activity and has been substituted for catecholamine measures. Also “i.e” should be replaced with “i.e.”
  • Answer:  As suggested alpha-amylase was defined as an indicator of the sympathetic adrenal medullary activity substituted for catecholamine measures; furthermore “i.e” has been replaced with  “i.e.”
  • Line 51: Remove capital “C” from “cortisol”.
    Answer: The capital C form cortisol has been removed
  • Line 52: remove space between “16, 17”.
    Answer: The space between “16, 17” has been removed
  • Line 52: This is simply due to the mechanism of synthesis of cortisol. Perhaps it would be appropriate to state that it is a more immediate indicator of stress reactivity compared with salivary cortisol.
  • Answer: As suggested, alpha-amylase has been stated that is a more immediate indicator of stress reactivity compared with salivary cortisol.
  • Line 54: define at first use of “SNS”.

Answer: the “SNS” has been defined as Sympathetic Nervous System

  • Line 84: The title of this section is Perhaps maintain this terminology instead of “subject”.
    Answer: As suggested, “subject” has been replaced with participants.

Page 3

-   Line 106: Alter to the Salivette Cortisol. It is assumed this was the type of collection tube      used? It is known that the Salivette cotton swab can influence the cortisol measures.

Answer: As suggested, has been better explained the type of collection tube used: “Salivette sampling device (Salivette, Sarstedt, Germany) were used to obtain saliva samples (>0.05 µl)”
-   Line 107: How long was the fast, the session was 12:00 – 14:00 so were they fasted after a breakfast meal or was if from the previous night.

Answer: As requested, it has been better explained that “Athletes were fasted from the breakfast meal (around seven o’clock) and they were asked not to rinse the mouth with water prior to sampling.”  
-  
Line 107: Can you confirm that each trial for each participant was completed at the same time of day i.e. if I completed the first trial at 13:00 all the other trials were at this time of the day.

Answer: We confirm, as already written in the paper, that each trial for each participant was completed at the same time of day

  • Line 126: Should this be Greenhouse-Geisser?
    Answer: Now, The Greenhouse-Geisser has been correctly written

Page 4

-    Line 147: It is not normally practice to present nanomoles as “nm”. I would alter these to nmol.

              Answer: As suggested, “nm” has been altered to nmol.

  • Figure 1: Indicate where the statistical differences were found

Answer: The statistical differences have been indicated in the Figure 1

Figure 1

Reviewer 2 Report

 Abstract: The authors’ start by characterizing TeamGym and describe the aim of the study is to investigate athletes’ stress-related changes during TeamGym training competition, but their conclusion discusses the sensitivity of one marker (alpha-amylase) to another (salivary cortisol) with a stressor. Thus, it is not in alignment. It is still unclear what the main purpose of this study was? Was it to test the responses and sensitivity of these two markers or was it to determine stress responses to this type of gymnastics (TeamGym)? In addition, it appears that the authors’ conclusion (Lines 26-28) is very generic and already well-known. 

Introduction:

The rationale for the study is weak and this reviewer is struggling with the novelty of this study. For example, several studies suggested that salivary alpha-amylase increases more rapidly than salivary cortisol to a stressor suggesting that it is a better index of stress (lines 50-52) and another study already demonstrated greater heart rate and cortisol responses during competition compared to training in gymnastics (lines 66-68) as well as other sports, thus what is the novelty of this study?

Line 49: typo

The authors’ hypothesize that the two major systems involved in responses to stress (sympathomedullary vs hypothalamic) would react differently to these two situations and that competition would impose a greater psycho-physiological load on the athletes when compared to training. The only novelty in this study it appears is using TeamGym as a stressor. The overall significance of this study is thus vague.

Is Figure 1 representing the same salivary amylase data as in table 2? It appears it is the same. A previous reviewer and editor requested a figure for cortisol for completeness. Because the results are really to compare these two responses, this reviewer agrees with the previous reviewer and editor and an additional figure is necessary. However, data should not be represented twice, so this reviewer recommends to eliminate Table 2.

Discussion: The novelty and impact of this manuscript are not clear to this reviewer. 

Author Response

Answer to Reviewer #2:

  • Abstract: The authors’ start by characterizing TeamGym and describe the aim of the study is to investigate athletes’ stress-related changes during TeamGym training competition, but their conclusion discusses the sensitivity of one marker (alpha-amylase) to another (salivary cortisol) with a stressor. Thus, it is not in alignment. It is still unclear what the main purpose of this study was? Was it to test the responses and sensitivity of these two markers or was it to determine stress responses to this type of gymnastics (TeamGym)? In addition, it appears that the authors’ conclusion (Lines 26-28) is very generic and already well-known. 

Answer: As requested, in the conclusion section our findings  have been better explained and linked with the main purpose; thus it was replaced: “TeamGym responses confirm other sports findings in stating that competition elicits higher stress response than training and suggest that salivary alpha-amylase is a more sensitive marker than cortisol to psycho-physiological stress also in gymnastics intermittent performance”.

  • Introduction: The rationale for the study is weak and this reviewer is struggling with the novelty of this study. For example, several studies suggested that salivary alpha-amylase increases more rapidly than salivary cortisol to a stressor suggesting that it is a better index of stress (lines 50-52) and another study already demonstrated greater heart rate and cortisol responses during competition compared to training in gymnastics (lines 66-68) as well as other sports, thus what is the novelty of this study?
  • The authors’ hypothesize that the two major systems involved in responses to stress (sympathomedullary vs hypothalamic) would react differently to these two situations and that competition would impose a greater psycho-physiological load on the athletes when compared to training. The only novelty in this study it appears is using TeamGym as a stressor. The overall significance of this study is thus vague.

Answer to number 2, 3: The rationale is to compare the different stress response in completely different situations: training and competition and therefore training should include more sessions similar to training (to increase the unpredictable stress of the competition in order to better train the athletes to face these situations). TeamGym, as gymnastics, depends on the perfection of movements therefore accustoming the athlete to a different type of stress can increase overall performance

  • Line 49: typo

Answer: The typo has been revisited

Answer:

  • Is Figure 1 representing the same salivary amylase data as in table 2? It appears it is the same. A previous reviewer and editor requested a figure for cortisol for completeness. Because the results are really to compare these two responses, this reviewer agrees with the previous reviewer and editor and an additional figure is necessary. However, data should not be represented twice, so this reviewer recommends to eliminate Table 2.

Answer: As requested, a figure of cortisol has been added and table 2 has been eliminated

  • Discussion: The novelty and impact of this manuscript are not clear to this reviewer. 

Answer: As required, a paragraph has been added to better explain the impact of this manuscript: “Only one study [21] demonstrated greater heart rate and cortisol responses during competition compared to training in gymnastics, but the results can be compared with some cautions to TeamGym performance: first of all, findings of Filaire and colleagues [21] concern peripubertal females gymnasts and describe their psychological and physical stress induced by training or competition which last one and a half minutes of continuous effort. No studies evaluated adult males and females gymnasts during training and official TeamGym competition [2,15] that require a longer trial than gymnastics one, with an intermittent effort on different sports equipment. Furthermore, the training method has not yet been well established in the TeamGym discipline; it is mainly borrowed from artistic gymnastics not always considering the technical and functional different performance demands.  Thus, it is reasonable to assume that the understanding TeamGym responses, in both settings, will provide valuable data regarding the magnitude of applied training loads and performance improvement [22].

Reviewer 3 Report

De Pero and colleagues have investigated athletes’ stress-related changes by comparing stress responses during training and official TeamGym competition througt assessing physiological measurements of stress, in particular hormonal and enzymatic responses, in adult gymnasts.

The introduction section is well written and result showing difference in changes of stress-related parameters during competition and training have good represented. The topic is definitely interesting, but the manuscript requires some revisions.

Major revisions

This study has limitations in its small sample size. If is not possible amlify the sample considered for the physiological measurements of stress, the authors should be better clarify in the discussion section that this study represents a preliminary investigation.

Furthermore, it should be add a discussion regarding future projects and possible training implication based on this evidence.

Minor revisions

  1. all abbreviation should be explained. Please check SNS in line 54
  2. the authors should provide a list of abbreviations

Author Response

Answer to Reviewer #3:

  • This study has limitations in its small sample size. If is not possible amlify the sample considered for the physiological measurements of stress, the authors should be better clarify in the discussion section that this study represents a preliminary investigation.
  • Furthermore, it should be add a discussion regarding future projects and possible training implication based on this evidence.

Answer:  As required, in the discussion section it has been better explained that: “As the small simple size, this study can be considered only a preliminary investigation: in the future in order to be complete, more information regarding psycho-physiological load (i.e. heart rate and blood lactate responses), mood changes [14] and anxiety [2] before training and competition could help coaches and physicians to individualize training and avoid non-functional overreaching.”

Furthermore, the following paragraph has been added to better explain possible training implication: “To achieve specific adaptations a training program must stress the systems that are exactly engaged during competition [22]; thus, TeamGym Coaches should aim to mimic competition demands during training sessions assessing internal stresses imposed by competition settings. TeamGym as gymnastics depends on the perfection of movements therefore accustoming the athlete to a different type of stress can increase overall performance. For example, coaches can implement training sessions by simulating competition with the same competition rest periods among trails, with athletes from other clubs or teams, or with the presence of external judges”

  • all abbreviation should be explained. Please check SNS in line 54

Answer:  As requested, all abbreviations have been explained. The SNS has been explained as “Sympathetic Nervous System”

Round 2

Reviewer 2 Report

The reviewer appreciates the authors providing more information/rationale for the study as well as the figure. The new text in its current form requires an editorial revision. The novelty of the current study is still unclear. The authors should consider their figure in the context of their statistical analysis. The authors are comparing the two responses and have observed an interaction. It is not clear why the authors are not representing the data to demonstrate the result of the statistical analysis.

Intro

Page 2, lines 49-50, alpha amylase is not a substitute for measuring catecholamines, it is a noninvasive estimate, please revise.

Lines 66-68: confusing sentence, needs revision

Lines 69-73: confusing sentence, revise for clarity. The reviewer is unclear of the authors’ point.

Indeed, the rest of the new text in red is similar. It appears that the authors are adding new text for a stronger rationale for this study, which is warranted and appreciated as this is important to gauge the novelty of the current study. This new text however, would benefit significantly from an editorial/language revision.

It is appreciated that the authors added the salivary cortisol figure. The figures however, should be consistent for ease of reviewing. With the type of graph chosen, it is easy to see why the authors switched the order of the dependent variables (training, rest and competition). If the point is to compare the two, could the authors use a different type of graph (bar or line) and include both the cortisol/alpha-amylase on one figure? This would be the best way to represent the data as it is the initial way the statistical analysis is set up with a two factor RM ANOVA with an interaction. The figures would then follow the analysis.

Lines 208-214 and 224-228 are confusing and would be strengthened by an editorial revision.

Author Response

Answer to Review:

The reviewer appreciates the authors providing more information/rationale for the study as well as the figure. The new text in its current form requires an editorial revision. The novelty of the current study is still unclear. The authors should consider their figure in the context of their statistical analysis. The authors are comparing the two responses and have observed an interaction. It is not clear why the authors are not representing the data to demonstrate the result of the statistical analysis.

Page 2, lines 49-50, alpha amylase is not a substitute for measuring catecholamines, it is a noninvasive estimate, please revise.

Answer: As suggested, alpha amylase as a non invasive estimate for meausring catecholamines has been corrected

Lines 66-68: confusing sentence, needs revision

Answer:As requested, lines 66-68 have been revised

Lines 69-73: confusing sentence, revise for clarity. The reviewer is unclear of the authors’ point.

Indeed, the rest of the new text in red is similar. It appears that the authors are adding new text for a stronger rationale for this study, which is warranted and appreciated as this is important to gauge the novelty of the current study. This new text however, would benefit significantly from an editorial/language revision.

Answer: As suggested, the new text has been revised.

It is appreciated that the authors added the salivary cortisol figure. The figures however, should be consistent for ease of reviewing. With the type of graph chosen, it is easy to see why the authors switched the order of the dependent variables (training, rest and competition). If the point is to compare the two, could the authors use a different type of graph (bar or line) and include both the cortisol/alpha-amylase on one figure? This would be the best way to represent the data as it is the initial way the statistical analysis is set up with a two factor RM ANOVA with an interaction. The figures would then follow the analysis.

Answer: As requested, figures have been changed using line graph. However, in our opinion, including both the cortisol/alpha-amylase on one figure, could make difficult for readers to understand the results, especially considering the high number of significance index.  

Lines 208-214 and 224-228 are confusing and would be strengthened by an editorial revision.

Answer: As requested, Lines 208-214 and 224-228 have been revised

Reviewer 3 Report

no suggestion

Author Response

Thank you for your opinion

This manuscript is a resubmission of an earlier submission. The following is a list of the peer review reports and author responses from that submission.

Round 1

Reviewer 1 Report

Dear Authors,

I congratulate you for the submitted manuscript and for its pertinence. Here are some comments:
The title is understandable and concise and it reflect the content.t
The resume includes: objectives, relevance of the theme and conclusions.
The introduction make clear the questions the authors want to answer and the objectives of the work.
The work was developed within an adequate theoretical framework.
The revised literature is sufficient for the depth of the theme presented, and the references cited current and relevant.
The conclusions presented are compatible with the topic under study.

A minor revision is suggested:
Materials and Methods
Participants
Line 74- it is suggested to indicate age, body mass and height by gender in order to better characterize the sample

results
placing the database on an open platform, for example, figshare data repository

Reviewer 2 Report

The authors provide a really interesting study about athletes’ stress-related changes during exercise.

Even if the topic is really interesting and only few studies exist, an N of 10 participants seems way too small to elicit meaningful effects. I therefore suggest the authors to 1.) provide a more comprehensive a-priori power analyses to detect a suitable sample size and then to 2) gain more data from more participants to see if current results can be replicated. (Why was f set to 0.35 and power only to 0.8? What approach was used – RMANOVA, within factors?  More information is needed here).

Due to a lack of theoretical foundation and discussion of results, I cannot suggest the study to be published in its current form. Below, I’ll provide some comments, which may be helpful for authors to re-structure their article before a follow-up submission.

General comments:
Why exactly was TeamGym chosen? What is it superiority compared to other (high intensity) sports? Why is it assumed to evoke best or most differentiate results in cortisol / amylase?

Line 52 ff.: here it becomes clear that the focus is on competition vs training loads. However, literature about general exercise effects on both stress-response systems are completely missing. Literature on competition effects is presented superficially and incomplete. The aim is somewhat “out of the blue”. Hence, a more thorough literature search and presentation is needed. Also, usage of TeamGym does still not become clear.

I suggest authors to perform a more thorough literature search as provided citations are quite fragmentary and draw an incomplete picture.

Specific comments:
Abstract

Line 19: “apparatus” is not clear here and needs to be clarified

Line 19: “during the recovery phase” seems to be misleading as I feel there is only one measurement within the recovery period

Line 20: I assume the protocol was tested twice in each participant? This does not become clear. So what is the aim of the study? To examine cortisol and amylase patterns, or to examine differences in trajectories between training and competition? What is the hypothesis underlying this assumption? Why should they be different?

Line 20/21: If the ANOVA did not reveal any differences, why are results shown for both, training and competition separately?

Line 21: I don’t understand the values provided here (e.g. -15=100.61+/-17.2U*ml-1)

Line 27: Cortisol results are kind of weird and should be further elaborated

Line 48: typo

Line 126 ff: Check correct citation of results

Libne 147: for the sake of completeness I suggest to also provide a Figure on cortisol responses

Line 153: However, this finding is not surprising, 153 considering that the training session was scheduled after returning from a 10 day training camp. Please elaborate on this. Why?